# The Success of a Screening Program Is Largely Dependent on Close Collaboration between the Laboratory and the Clinical Follow-Up of the Patients

**DOI:** 10.3390/ijns6030068

**Published:** 2020-08-26

**Authors:** Svetlana Lajic, Leif Karlsson, Rolf H. Zetterström, Henrik Falhammar, Anna Nordenström

**Affiliations:** 1Department of Women’s and Children’s Health, Karolinska Institutet, SE-17176 Stockholm, Sweden; Svetlana.Lajic@ki.se (S.L.); Leif.Karlsson@ki.se (L.K.); 2Pediatric Endocrinology Unit, Astrid Lindgren Children’s Hospital, Karolinska University Hospital, SE-17176 Stockholm, Sweden; 3Center for Inherited Metabolic Diseases, Karolinska University Hospital, SE-17176 Stockholm, Sweden; Rolf.Zetterstrom@ki.se; 4Department of Molecular Medicine and Surgery, Karolinska Institutet, SE-17176 Stockholm, Sweden; Henrik.Falhammar@ki.se; 5Department of Endocrinology, Metabolism and Diabetes, Karolinska University Hospital, SE-17176 Stockholm, Sweden

**Keywords:** neonatal screening, congenital adrenal hyperplasia, CAH, 21-hydroxylase deficiency, long-term outcome

## Abstract

Neonatal screening for congenital adrenal hyperplasia due to 21-hydroxylase deficiency is now performed in an increasing number of countries all over the world. The main goal of the screening is to achieve early diagnosis and treatment in order to prevent neonatal salt-crisis and death. The screening laboratory can also play an important role in increasing the general awareness of the disease and act as the source of information and education for clinicians to facilitate improved initial care, ensure prompt and correct glucocorticoid dosing to optimize the long-term outcome for the patients. A National CAH Registry and *CYP21A2* genotyping provide valuable information both for evaluating the screening program and the clinical outcome. The Swedish experience is described.

## 1. Introduction

Congenital adrenal hyperplasia (CAH) due to 21-hydroxylase deficiency (21OHD) results in cortisol and aldosterone deficiency. The compensatory increase in ACTH production and concomitant androgen excess is present already in utero and results in virilization of external genitalia in fetuses with a 46, XX karyotype [1]. Before neonatal screening was implemented there was a marked female preponderance among patients with CAH since boys with CAH were diagnosed less often due to a lack of obvious clinical symptoms prior to developing a life-threating salt-losing adrenal crisis in the neonatal period [2]. Lifelong treatment with replacement doses of glucocorticoids and fludrocortisone is required, and during the first year of life additional sodium chloride [1]. Neonatal screening for CAH is now performed in an increasing number of countries throughout the world [3]. The screening was initiated in Sweden in 1986 [4]. The main goal of the screening program is to prevent adrenal salt-crisis and death in the neonatal period [5]. As a secondary benefit, children with less severe forms of CAH diagnosed early may escape early androgen symptoms and instead achieve normal growth and development. The care of individuals with CAH has developed over the past 50 years and the focus is nowadays not only on identification and diagnosis to save lives but the aim is also to improve long-term outcomes and quality of life for the patients [6].

The combination of an efficient neonatal screening program and an optimal treatment and follow-up is key to achieving best possible long-term patient outcomes. Here we describe the Swedish experience and the outcomes and benefit of a close collaboration between the screening laboratory, clinical care, and follow-up.

### 1.1. Screening

There is one national screening laboratory in Sweden managing more than 100,000 samples per year. Newborn screening is not mandatory in Sweden, but virtually all newborns are screened. The families are given written information at the time the sample is collected and an opt out procedure is employed. Since 2010, the filter paper samples have been collected as soon as possible after 48 h, but were previously collected on day 3–5 [5]. The concentration of 17-hydroxyprogesterone (17OHP) is measured using Genetic Screening Processor (GSP) instruments (Perkin Elmer, Waltham, MA, USA). Gestational age related cut-off levels are used. The cut-off levels have been gradually adjusted over time. At present the cut-off level for full term infants born in or after gestational week (GW) 37 is 60 nmol/L (assuming a hematocrit of 50% in the blood samples). Infants born in GW 35–36 have a cut-off level of 100 nmol/L, and preterm babies born in or before GW 34 have a cut-off level of 350 nmol/L [7].

### 1.2. Genetics

The genotype-phenotype correlation for 21OHD is well described [8]. Although more than 200 different mutations have been identified today, there is a limited number of mutations that make up more than 90% of the alleles described in patients worldwide [9]. The patients can be divided into four genotype groups based on the severity of the milder allele in compound heterozygous patients: null, I2 splice, I172N, and non-classic CAH. Generally, the null and I2 splice groups are associated with the SW phenotype, I172N with SV CAH, and V281L with NC CAH. P30L results in a phenotype between SV and NC, but was in this study defined as non-classical. A detailed description of all the different mutations in the Swedish cohort has been described elsewhere [6].

## 2. Short Term Perspective

### 2.1. Clinical Investigation and Diagnostic Work Up

All screening results above the cut-off level are considered positive screening results [5]. A physician at the laboratory calls out the result to a pediatrician in charge at the local hospital where the child was born, or in the case of preterm infants, to the neonatal ward where the child is presently cared for (see Figure 1). The local physician in charge can thereby receive information about required investigations and treatment and ask questions. The level of 17OHP gives important prognostic information [10]. Advice regarding the dose of hydrocortisone, fludrocortisone, and the likelihood of developing an adrenal salt-crisis and whether the child will require additional sodium chloride can be given. When the level of 17OHP in the screening sample is high and the child is suspected to have the salt wasting form of CAH [10], the urgency is stressed and clinical advice is given about treatment. Consideration of screening 17OHP levels is an important step which provides the opportunity to give detailed recommendations on which additional blood samples to take and whether treatment should be started immediately. Information can be individually tailored to the suspected severity of disease for the specific child. Equally important is the information that a child with a moderately elevated 17OHP, and hence a low risk of salt-loss, can be called in for less urgent assessment the following day. Treatment should in those cases await the results of the follow-up blood tests unless the child has developed symptoms.

It is not uncommon that the child is already admitted to the hospital or followed-up because of weight loss or insufficient weight gain when the result of the screening is ready. Hypoglycemia during the first few days of life may also be information that strengthens the suspicion of CAH. The initial symptoms to look for in the neonate with a positive screening test are insufficient weight gain, vomiting, lethargy, hyperpigmentation, clitoral enlargement/penis size, presence or absence of palpable testes, and increased anogenital distance. Analysis of electrolytes, 17OHP, cortisol, a second screening sample on a filter paper card and possibly other hormone analyses depending on the clinical symptoms should be performed. *CYP21A2* mutation analysis gives important prognostic information regarding disease severity and is also informative in investigations of cases where the biochemical diagnostics continue to be unclear.

In cases when the sex of the child is unclear the screening laboratory can perform the screening analysis of 17OHP earlier than at 48 h, and thereby sometimes shorten the time to diagnosis.

### 2.2. Medical Treatment

The initial hydrocortisone dose given to the child with a newly detected SW form has to be adjusted depending on the clinical situation. If the child has elevated potassium and decreased sodium, treatment should start immediately. Intravenous glucose infusion containing sodium and intravenous hydrocortisone should be given promptly. The initial hydrocortisone bolus dose of 5 mg/kg is followed by 25 mg per 24 h, as a continuous infusion or divided in 3 or 4 doses in our center. The difficulty is often to evaluate if a child is clinically affected when the electrolytes are within the normal range. Our advice, if the diagnosis is likely, considering the level of 17OHP and other clinical symptoms, is to start the treatment with 5 mg × 3 of hydrocortisone orally or intravenously for 2–3 days and then taper down to 2.5 mg × 3 and finally 1 mg × 3. Addition of fludrocortisone and sodium chloride can be given orally if the 17OHP level in the screening is clearly elevated or when the clinical picture is clear and the parents have learned how to give the medication. It may be as important not to over-treat as to avoid an adrenal salt-crises also in the neonatal period.

The *CYP21A2* genotype is determined for virtually all patients in Sweden. The genotype-phenotype correlation for 21OHD is good and genotyping therefore gives valuable support in treatment decisions. Genotyping is also helpful in unclear cases for whom NC CAH may be suspected. However, not all individuals with NC CAH will benefit from starting treatment early [11]. Once a patient is on continuous replacement therapy, it will lead to a decrease in the endogenous glucocorticoid and mineralocorticoid synthesis and put the patient at risk of salt-crises in the event of a stressful situation [11].

## 3. Long-Term Follow-Up and Perspective

Evaluation of the outcome of the screening program, positive predictive value (PPV) and sensitivity, as well as the outcome for the patients requires follow-up of all patients diagnosed via the screening and patients with a late diagnosis missed by the screening [5]. In Sweden, this is facilitated by the use of a National CAH Registry comprising more than 700 patients today [6]. The severity of CAH among the missed or late diagnosed patients can be determined by *CYP21A2* genotyping. In the registry the *CYP21A2* genotype is known for more than 80% of the patients.

The registry has also made epidemiological studies possible. The patients were anonymized and linked to data of national population-based registers for mortality, somatic, and psychiatric diagnoses in Sweden. One hundred age- and sex-matched controls were selected per patient. In addition, we were able to assess the *CYP21A2* genotype separately (null, I2 splice, I172N, P30L, and NC CAH), as well as outcomes before and after the introduction of the national neonatal screening in 1986.

### 3.1. Mortality

In two epidemiological studies, one from Sweden and one from the UK, an increased mortality rate was found in patients with CAH (hazard ratio 3–5) with death occurring 6.5–18 years earlier than for controls [12,13]. We were able to show that the neonatal and first year mortality virtually disappeared with the implementation of the neonatal screening for CAH. Interestingly, it was not only the mortality among boys with CAH that was prevented [6]. More girls with the most severe SW form survived with screening, which can be explained by the screening identifying girls who would otherwise have been diagnosed with hypospadias and not have received hydrocortisone treatment. We could also show that the mortality among adults continues to be elevated with a hazard ratio of 2.3–3 in men and women respectively. In almost 50% of the cases the death was related to salt-crises, also among adults. This finding is contradictory to what has been previously generally believed, that the adrenal salt-crises is mainly a risk among children and not a large problem among adults. Next after adrenal crisis, cardiovascular death was the most common cause of death [12] and there was also a documented increased cardiovascular morbidity [14].

### 3.2. Cardiovascular and Metabolic Risk

Glucocorticoid replacement and androgen control impact the cardiovascular and metabolic risk in patients with CAH [14,15,16]. Obesity and hypertension, even in young people, may be caused by long-term glucocorticoid replacement [17,18]. Thus, many studies on CAH have suggested an increased cardiovascular and metabolic risk, including insulin resistance [19,20,21]. However, only occasional studies have been able to show an increased frequency of established cardiovascular disease or diabetes [14,16]. This could be expected since very few studies have included patients with CAH above the age of 50 years when cardiovascular disease and diabetes usually appear. Different genotypes may have different risk for cardiovascular events [14]. In the epidemiological study of the entire Swedish population and in all patients with CAH (*n* = 588) the risk for any cardiometabolic disease, any cardiovascular disease, obesity, diabetes, obstructive sleep apnea, dyslipidemia, hypertension, atrial fibrillation, and venous thromboembolism was significantly increased compared to matched controls (*n* = 58,800) [14]. Cardiovascular and metabolic issues still occurred in those born after the introduction of neonatal screening in 1986 in Sweden, especially any cardiovascular and metabolic disorder, obesity, and hypertension [14]. However, other studies have indicated that a late diagnosis of CAH may be associated with increased cardiometabolic risk [16,20]. Thus, early diagnosis and regular monitoring of cardiometabolic risk in patients with CAH is important.

### 3.3. Fertility Issues

Fertility has been reported to be impaired in both females and males with CAH [9,22]. In women high androgen and 17OHP concentrations lead to menstrual irregularities and anovulation [1]. Continuous high progesterone concentrations result in a contraceptive effect, but optimizing glucocorticoid replacement can normalize the hormonal situation and improve fertility [23]. However, decreased sexual activity, higher sexual distress, higher prevalence of homosexuality or bisexuality, and disinterest in pursuing motherhood most likely play a role in the decreased fertility rate [24,25]. Those with a more severe genotype/phenotype have a lower fertility rate [22,24]. Fertility in males with CAH is impaired [26,27], mainly due to hyper- or hypogonadotropic hypogonadism and testicular adrenal rest tumors (TARTs) [28]. These tumors were reported in up to 86% of all adult males with CAH [29]. In the Swedish epidemiological study including 221 males with CAH only those born before the neonatal screening had impaired fertility while those born after the introduction of screening had normalized fertility compared to same age controls [27]. One can speculate if the mini-puberty or an imbalance in FSH, LH, and gonadal synthesis of testosterone or other steroid hormones during the first year of life may be of importance [30]. It is unclear if the same is true for females with CAH but fertility rates have improved over the years and may now slowly approach that of the general population [31]. If the improved fertility is due to better and individualized fertility therapy, better general management of CAH or the introduction of neonatal screening is unclear. Thus, early diagnosis may improve fertility, at least in males with CAH.

### 3.4. Stress Vulnerability and Psychiatric Diagnoses

Stress vulnerability seems to be increased in patients with CAH and has been suggested to be one of the causes of increased sick leave and disability pension in CAH compared to matched controls [22]. Only a few investigations on psychiatric diagnosis in CAH have been reported. Psychiatric diseases seem more common [13,16,32,33], especially depression [13,32,33,34], alcohol misuse [32,33] and suicidality [16,32], which can be interpreted as stress related. Around 10% of the mortality in CAH was caused by suicide [12] but in males with CAH born after the introduction of the neonatal screening no increase in any psychiatric disorder or suicidality were seen [32]. However, in females with CAH a similar over representation in any psychiatric disorder were seen before and after the introduction of the neonatal screening, even though the spectrum of psychiatric diagnosis was slightly different [33]. Females with the null genotype born before the introduction of the neonatal screening were more likely to be diagnosed with ADHD [33]. The explanations may be increased androgen exposure, repeated episodes of hypoglycemia or salt-crises, all more prevalent before the introduction of neonatal screening. Moreover, males diagnosed late had more depressive symptoms and lower self-control compared to controls [35]. Early exposure to elevated androgen levels or supra-physiologic glucocorticoid doses may have secondary effects on the function of the pituitary–adrenal-axis later in life, contributing to stress vulnerability [36]. Hence, a later diagnosis may increase the risk of a psychiatric diagnosis.

### 3.5. Cognition

It is known that the brain is sensitive to exposure to high levels of cortisol and other glucocorticoids and that it may be more vulnerable early in the development. The treatment in CAH is life long and it is a continuous process to adjust the dose of glucocorticoid throughout childhood and adult life. Worldwide studies have repeatedly shown that individuals with CAH have negatively affected cognition. However, the results from studies investigating cognitive outcome in patients with CAH have been disparate. In some studies, adults with CAH have a lower full-scale IQ compared to controls [37,38,39,40], but other studies show a normal general intellectual capacity, irrespective of age [41]. While results differ between studies regarding the full-scale IQ, an observation that seems to be consistent is that patients with CAH often have an impaired working memory performance [42,43,44,45]. The effects on working memory were seen as early as during childhood in patients that did not undergo neonatal screening for CAH [43].

Interestingly, in a Swedish cohort of children with CAH, with good metabolic control, that was identified through neonatal screening and treated early with a three- or four-dose regimen with hydrocortisone we could not see any negative impact on executive functioning or memory [46]. This emphasizes the importance of an early diagnosis, before the development of an adrenal crisis, and adequate glucocorticoid dosing and good adherence to therapy [46].

More recent observational studies from the UK, USA, and Sweden regarding brain morphology in patients with CAH have revealed that patients show reductions in cortical and limbic regions of the brain that are important for the working memory [45,47,48]. This new evidence points to the fact that alterations in brain structure and possibly also in the functional organization of the brain may underlie the cognitive changes [45,47,48]. In the UK study [45], patients with CAH showed reductions in volumes of the right hippocampus, left amygdala, bilateral thalamus, cerebellum, and brainstem. The study cohort included only women (age range 18–50 years), most patients were treated with hydrocortisone, and 40% with prednisolone. The majority of the patients had SW CAH and 74% of the patients had a null genotype [45].

The American cohort [48] comprised children (age range 8–18 years) and analyzed the MRI data with a focus on the prefrontal cortex and regions of the hippocampus and the amygdala. The patients almost exclusively had SW CAH and approximately half of the cohort was diagnosed through neonatal screening. Patients were treated either with hydrocortisone, prednisolone, or dexamethasone. Also here, reductions in the volumes of the prefrontal cortex, amygdala, and hippocampus were observed [48].

The Swedish study [47] included both women and men in a large cohort of adolescents and young adults with CAH (age range 16–32 years). Most patients were treated with hydrocortisone and the majority was diagnosed through the Swedish neonatal screening program for CAH. The authors could not identify structural disturbances in limbic structures but changes were observed in the prefrontal, parietal, and superior cortex, regions important for working memory functioning. Interestingly, there was a positive association between the grey matter structures and the nucleus precuneus with working memory performance [47].

Imbalances in hormonal levels such as over- or undertreatment with glucocorticoids or the difficulty in mimicking the circadian and ultradian cortisol rhythm, repeated episodes of hypoglycemia, or salt-losing crises may affect brain structures and the cognitive abilities in patients with CAH [37,38,39,40,42,43,44,46]. Given this, optimal glucocorticoid treatment is a key factor in improving long term cognitive outcomes in patients with CAH.

Neurons within the amygdala, hippocampus, and the prefrontal cortex express both mineralocorticoid receptor and glucocorticoid receptor at high levels, and the glucocorticoid receptor is also widely expressed throughout the brain [49,50,51]. These areas, important for executive functioning, emotional regulation, and memory [52,53,54] have been shown to be vulnerable to high levels of glucocorticoids. In addition, the severity of CAH has also been shown to affect the cognitive outcome possibly through differences in androgen exposure, hypoglycemia, salt-losing crises, and differences in treatment regimen [43,44]. The differences in patient outcome most likely is the product of a combination of the previously mentioned factors and the choice of treatment such as type of glucocorticoid [40,45,46].

The vulnerability of the brain in the neonatal period and early in life makes it important to detect the patients as early as possible to avoid adrenal salt-crises, and possibly hypoglycemia before start of treatment. It is possible that, in addition, avoiding treatment with extremely high stress-doses of glucocorticoid in the neonatal period may be an important factor. Early diagnosis and treatment may also be important for psychological factors, vulnerability to stress, and cardiovascular/metabolic morbidity over the course of the lifetime. Close follow-up and adjustment of the replacement doses to mimic the circadian rhythm, avoid salt crises and hypoglycemia, and at the same time avoid overtreatment all through childhood is of course imperative but at times a challenging task.

## 4. Conclusions

Improvements in the overall outcome for patients with CAH require a close collaboration between the screening laboratory and the clinicians. *CYP21A2* genotyping is instrumental in the follow-up and evaluation of the screening program and gives valuable guidance in the clinical care of the patients. The screening program increases the general awareness of the disease and the knowledge about treatment. The screening lab can also act as the source of information and education for clinicians especially in a large country with a relatively small population like Sweden with few specialized centers. This facilitates improved initial care, ensures prompt and correct glucocorticoid dosing, and avoids overtreatment and its potential negative effects on the brain and development. From the current evidence, it seems that implementation of neonatal screening and avoiding long-acting and high doses of glucocorticoids complemented with a good clinical management over time is key for improving long-term outcome for the patients with CAH.

## Figures and Tables

**Figure 1 IJNS-06-00068-f001:**
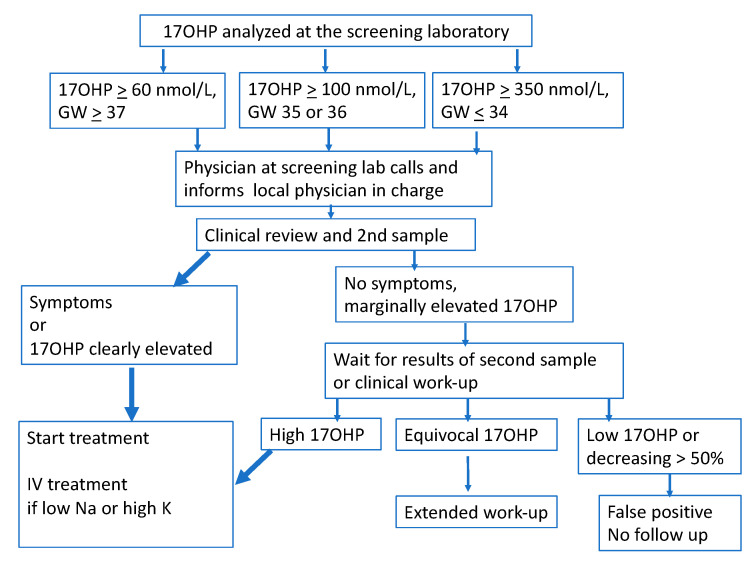
Flow chart for the neonatal screening, 17OHP cut-off levels, information, and investigations.

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
