# Peer review of "The Success of a Screening Program Is Largely Dependent on Close Collaboration between the Laboratory and the Clinical Follow-Up of the Patients"

_2409-515X, 2020, doi:10.3390/ijns6030068_

Round 1

Reviewer 1 Report

The paper successfully argues for the benefit of close collaboration between the screening lab and clinical services. There is a point around 17OHP levels that need clarification, and otherwise some minor English language corrections.

Line 51; "and follow-up is the key to achieve best possible outcome for the patients in a longer perspective." Suggest re-phrasing to "and follow-up is key to achieving best possible long-term patient outcomes."

Line 59; "Since 2010 the filter paper samples are collected as soon as possible after 48 hours, " suggest "Since 2010 the filter paper samples have been collected as soon as possible after 48 hours."

Line 63; Are the screening 17OHP levels whole blood or corrected to serum? The provision of the assumed Hct suggests that these are serum levels. This should be clarified.

Figure 1 "Undecisive 17OHP"  Suggest describing this as "Equivocal." 

Line 85; Given that this relates to a screening rather than diagnostic test (which are presumably performed  even if treatment starts before results are available), suggest rephrasing as suspected or likely to have eg, "the child is suspected to have the salt losing form of CAH."

Lines 86-91; Suggest re-phrasing. "Consideration of screening 17OHP levels is an important step which provides the opportunity to give detailed recommendations on which additional blood samples to take and whether treatment should be started immediately. Information can be individual tailored to the suspected severity of disease for the specific child. Equally important is the information that a child with a moderately elevated 17OHP, and hence a low risk of salt loss, can be called in for less urgent assessment the following day."

Line 118; Suggest "for 2-3 days" rather than "during 2-3 days" and 2.5 mg should have a point rather than comma.

Line 129; Suggest replacing "true outcome" with "performance" or just referring to "screening program outcomes"

Line 151; Suggest "not have received hydrocortisone."

Line 184; Suggest "disinterest in pursuing motherhood."

Line 276; Suggest "and possibly hypoglycaemia, before the start of treatment."

Author Response

We thank the reviewer for the suggested English language corrections. They have all been implemented.

Line 63; Are the screening 17OHP levels whole blood or corrected to serum? The provision of the assumed Hct suggests that these are serum levels. This should be clarified.

Response: Since the start of the screening in Sweden a calculated plasma level assuming a hematocrit of 50% in all samples has been used. The measurements are performed in blood from the DBS. We have tried to make this more clear by adding  - in the blood samples

Reviewer 2 Report

There are multiple English grammar edits.  I have outlined a majority of them below. However the paper would benefit from a writer editing for grammar throughout.

Title:  Would recommend making the following edit:  “…largely dependent on”

Be consistent with abbreviation of 17-hydroxyprogesterone throughout the manuscript. E.g. Figure 1 legend.

Introduction, lines 37-38: Would recommend “…. boys with CAH were diagnosed less often due to a lack of…”

Line 38: “life-threatening”. Also, it seems more conventional to refer to a ‘salt crisis’ as “salt-wasting adrenal crisis” or “salt-wasting crisis” in the CAH literature.

Line 45: “… have normal growth and development. The care of individuals with CAH…”

(Line 43-45 is a bulky sentence, beginning with ‘A secondary benefit’ and should be edited)

Line 50: “… screening program and optimal treatment and follow-up is the key to achieving the best…”

Line 53: “… and the outcomes and benefit of a close collaboration between the laboratory, clinical care, and follow-up”.

Line 61: misspelled instruments. Define GSP.

Line 76: Change to “Short term perspective”

Line 88: repeats information twice in the sentence. Would edit.

Line 105: should be “continues” instead of continuous. That sentence beginning on line 103 requires some editing. "... analysis gives important prognostic information regarding disease severity and is informative in cases where the biochemical diagnostics continue to be unclear."

Line 130: “longer-term perspective”’.  But actually, could consider renaming as “Long-term follow-up and perspective”

Line 150: “screening and identifying of girls who would otherwise…”

Line 152: “continues to be elevated…”

Lines 192-193: awkwardly written. Perhaps “It is unclear if the same is true for females with CAH, but fertility rates have improved…”

Line 265: "outcomes"

Line 287:  “Improvements in the overall outcome for patients with CAH require a…”

Lines 293- : “This facilitates improved initial care, ensures prompt and correct glucocorticoid dosing, and avoids overtreatment and its potential negative effects on the brain and development”.

Conclusion, final sentence – is a long sentence that will require editing.

This is an interesting paper outlining the Swedish experience with newborn screening, as well as its relationship to follow-up data in the way of adverse outcomes. As the country has a National CAH Registry to provide follow-up data, and a strong genotype-phenotype experience, it makes the paper interesting to read. 

The authors state a good case for the early diagnosis of classic CAH via newborn screen, discussing their experience and that of other centers, including:  significant improvement in mortality in children, adverse cardiometabolic outcomes potentially being associated with a late diagnosis of CAH, and fertility and mental health potentially being improved in males. They also review brain structure and function, and the potential for insults early in life to affect the vulnerable neonatal brain (salt wasting crises, hypoglycemia, high glucocorticoid doses). Also of interest is the discussion about identifying NC CAH patients via genotyping. However, there is not much direct discussion about genotype and the longer-term outcomes. 

One distinction that is unclear is the role of the screening program vs. the screening laboratory. At times, the two almost seem to be referred to interchangeably. I am wondering if it is the screening laboratory or the screening program that has the close collaboration with the clinicians. Although it depends on the setup for Sweden. The physicians call from the lab to the clinicians, but sometime is the screening program contacting the clinician? Or perhaps that is referred to in regards to other countries?

It is unclear and could perhaps be explained further in the section named Screening. Is it because Sweden is a large country with a relatively small population that there is not a screening program (and instead just the screening laboratory)?

As well, could it be further explained why ‘most patients in Sweden’ have a genotype determined, but not all? (Line 123)

Author Response

There are multiple English grammar edits.  I have outlined a majority of them below. However the paper would benefit from a writer editing for grammar throughout.

Title:  Would recommend making the following edit:  “…largely dependent on”

Response: Thank you, this has been changed

Be consistent with abbreviation of 17-hydroxyprogesterone throughout the manuscript. E.g. Figure 1 legend.

Introduction, lines 37-38: Would recommend “…. boys with CAH were diagnosed less often due to a lack of…”

Line 45: “… have normal growth and development. The care of individuals with CAH…”

(Line 43-45 is a bulky sentence, beginning with ‘A secondary benefit’ and should be edited)

Line 50: “… screening program and optimal treatment and follow-up is the key to achieving the best…”

Line 53: “… and the outcomes and benefit of a close collaboration between the laboratory, clinical care, and follow-up”.

Line 61: misspelled instruments. Define GSP. The abbreviation has been spelled out.

Line 76: Change to “Short term perspective”

Line 88: repeats information twice in the sentence. Would edit.

Line 105: should be “continues” instead of continuous. That sentence beginning on line 103 requires some editing. "... analysis gives important prognostic information regarding disease severity and is informative in cases where the biochemical diagnostics continue to be unclear."

Line 130: “longer-term perspective”’.  But actually, could consider renaming as “Long-term follow-up and perspective”

Line 150: “screening and identifying of girls who would otherwise…”

Line 152: “continues to be elevated…”

Lines 192-193: awkwardly written. Perhaps “It is unclear if the same is true for females with CAH, but fertility rates have improved…”

Line 265: "outcomes"

Line 287:  “Improvements in the overall outcome for patients with CAH require a…”

Lines 293- : “This facilitates improved initial care, ensures prompt and correct glucocorticoid dosing, and avoids overtreatment and its potential negative effects on the brain and development”.

Conclusion, final sentence – is a long sentence that will require editing. 

Response: The sentence has been shortened and changed.

Response: Thank you the suggested English language corrections. They have all been implemented

Line 38: “life-threatening”. Also, it seems more conventional to refer to a ‘salt crisis’ as “salt-wasting adrenal crisis” or “salt-wasting crisis” in the CAH literature.

This is an interesting paper outlining the Swedish experience with newborn screening, as well as its relationship to follow-up data in the way of adverse outcomes. As the country has a National CAH Registry to provide follow-up data, and a strong genotype-phenotype experience, it makes the paper interesting to read. 

Response: Thank you.

The authors state a good case for the early diagnosis of classic CAH via newborn screen, discussing their experience and that of other centers, including:  significant improvement in mortality in children, adverse cardiometabolic outcomes potentially being associated with a late diagnosis of CAH, and fertility and mental health potentially being improved in males. They also review brain structure and function, and the potential for insults early in life to affect the vulnerable neonatal brain (salt wasting crises, hypoglycemia, high glucocorticoid doses). Also of interest is the discussion about identifying NC CAH patients via genotyping. However, there is not much direct discussion about genotype and the longer-term outcomes.

Response: A more detailed discussion and assessment of the genotypes in relation to loger-term outcomes can be found in our previous publications on the national cohort, and was not the focus of this manuscript. In addition, we feel that a more extensive discussion about the genotype and long term out-comes is not possible within the frames of this manuscript.

One distinction that is unclear is the role of the screening program vs. the screening laboratory. At times, the two almost seem to be referred to interchangeably. I am wondering if it is the screening laboratory or the screening program that has the close collaboration with the clinicians. Although it depends on the setup for Sweden. The physicians call from the lab to the clinicians, but sometime is the screening program contacting the clinician? Or perhaps that is referred to in regards to other countries?

It is unclear and could perhaps be explained further in the section named Screening. Is it because Sweden is a large country with a relatively small population that there is not a screening program (and instead just the screening laboratory)?

Response: The person hired at the screening laboratory who calls out the results is a pediatrician or rather a pediatric endocrinologist. In the text the screening laboratory and the screening program are used interchangeably, the laboratory is running the screening program. We have tried to make this more clear in the revised manuscript.

As well, could it be further explained why ‘most patients in Sweden’ have a genotype determined, but not all? (Line 123)

Response: It is the clinician responsible for the patient who decides about the genetic investigation. The screening laboratory is not performing the genetic analysis. For older patients mutation analysis has not always been performed, although more than 80% of the patients in Swede have been genotyped (Gidlöf 2013 in Lancet D&E). We show in another manuscript for the special issue that 100 % of the diagnosed patients have been genotyped since 2011. We have rephrased the sentence in order not to create confusion “The CYP21A2 genotype is determined for virtually all patients in Sweden.”